# Trait Mindfulness, Self-Compassion, and Self-Talk: A Correlational Analysis of Young Adults

**DOI:** 10.3390/bs12090300

**Published:** 2022-08-23

**Authors:** Jocelyn Grzybowski, Thomas M. Brinthaupt

**Affiliations:** Department of Psychology, Middle Tennessee State University, Murfreesboro, TN 37132, USA

**Keywords:** self-talk, self-compassion, trait mindfulness, mindfulness practice

## Abstract

This research explores the relationships between trait mindfulness, self-compassion, self-talk frequency, and experience with mindful practice. We expected to find that positive self-talk would be positively related to mindfulness and self-compassion, and negative self-talk would be negatively related to these variables. Participants (*N* = 342) were recruited through a university research pool, as well as via social media posting. The participants completed two measures of trait mindfulness (the 15-item Five Facet Mindfulness Questionnaire and the Trait Toronto Mindfulness Scale), two measures of self-talk (the Self-Talk Scale and the Automatic Thoughts Questionnaire—Revised), and the Self-Compassion Scale short form. The results showed moderate positive correlations between (1) positive self-talk and trait mindfulness and (2) positive self-talk and self-compassion. A significant negative correlation also emerged between negative self-talk and trait mindfulness. Additional analyses indicated no moderating effects of mindfulness experience on self-talk or self-compassion in predicting trait mindfulness. We discuss implications for the significance of the relationship between self-talk and mindfulness for the effective implementation in future treatment methodologies.

## 1. Introduction

Mindfulness-based cognitive behavioral interventions have become increasingly popular in recent years. Contemporary cognitive behavioral psychology frequently adopts mindfulness as a tool to help patients combat affective unbalance and maladaptive behaviors through insight and awareness [1] alongside management of self-critical self-talk and other ruminative thought behaviors [2]. Mindfulness-based interventions (MBIs) have shown successful treatment outcomes and symptom management across several physical disorders, mental disorders, and crisis states [3,4,5,6]. Traditionally, MBIs aid in the development of metacognitive skills linked with mindful attention to assist in the remediation of negative psychological symptoms. In particular, mindful meditation and a self-compassionate approach have been associated with improved psychological wellbeing and self-regulatory behavioral patterns [7,8]. It has been theorized that a fleshed-out model of mindfulness may involve a unilateral decrease in intrapersonal communication more generally [9].

Mindful practice engages several cognitive mechanisms that result in the reduction in negative symptoms associated with psychopathology, as indicated by improvements in physical health, psychological health, general wellbeing, self-regulation, and executive functioning [10,11,12]. Despite the proven efficacy of these methods of treatment, questions continue to arise regarding the influence of key processing variables which could inhibit or assure positive treatment outcomes when implementing various psychological interventions. Three such variables, self-talk, self-compassion, and mindfulness, are the focus of the present research. In this study, we examine the relationships among these variables. In particular, the study’s objectives are to examine how positive and negative self-talk are related to trait mindfulness and self-compassion. We first review the literature pertaining to these concepts, after which we highlight likely relationships among them.

## 2. Literature Review

### 2.1. Trait Mindfulness Facets

Adapted from centuries-old Buddhist ideology and Eastern tradition, mindfulness describes a developed metacognitive awareness which assists us on our journey toward the end of suffering [13]. Within the field of psychology, however, the definition of mindfulness has been expanded beyond a means to end suffering. Psychologists tend to agree that mindfulness involves deliberate attention and awareness in the present moment [1,12,14]. There is also a shared consensus that mindfulness has both state (or situational) and trait (or dispositional) attributes [15]. As we did not perform a direct intervention, we focus on dispositional mindfulness rather than any momentarily induced mindful state.

Researchers have noted the trait mindfulness is a multidimensional construct. For example, Baer et al.’s [16] mindfulness measure includes facets related to people’s tendencies to observe and describe their surroundings and experiences, acting with or without awareness, and being non-judgmental about and non-reactive to various inner experiences. Davis et al.’s [17] scale includes two facets related to people’s tendencies to distance themselves from everyday experiences and to be curious about and aware of things that are happening to them.

Trait mindfulness research has identified several significant correlations of emotional balance and cognitive performance as they relate to overall psychological wellbeing [12]. In a meta-analytical study of trait mindfulness studies, Mesmer-Magnus et al. [18] found that overall psychological wellbeing, as well as a number of other indicators of psychological health, were positively correlated with trait mindfulness. Though the authors’ primary focus was to gauge effects on occupational outcomes, other metacognitive factors affecting daily functioning, such as positive affect, psychological wellbeing, psychological flexibility, confidence, and general life-satisfaction, were also positively correlated with trait mindfulness.

Trait mindfulness and other self-oriented mental health strategies, such as self-compassion, have been shown to directly mediate psychological health outcomes in mindfulness-based intervention programs [19]. Additionally, Mesmer-Magnus et al. [18] noted negative correlations between mindfulness and certain psychological experiences, such as perceived life stress, negative emotions, depression, and anxiety. This research supports a connection between mindfulness-based interventions and effective treatment of mood and anxiety disorders.

### 2.2. Self-Compassion Characteristics

Often referred to as an attitude developed within mindful practice, self-compassion involves self-kindness in the place of self-judgment, acquired through introspection and the consideration of the emotional tone one adopts toward speaking with themselves [20,21]. Research suggests that a self-compassionate approach may function as an antidote for maladaptive intrapersonal communication behaviors, such as depressive rumination and anxious worrying [22]. For example, researchers find that higher levels of self-compassion are associated with lower levels of negative self-evaluation, such as self-criticism, shame, and self-blame [23,24]. Research also suggests that higher self-compassion is associated with an increased ability to deal with personal troubles or failures [25,26]. Self-compassion is theorized to allow for effective self-regulation and behavioral adjustment via self-monitoring to prioritize adaptive, self-empathetic thought processes and communication [20,27].

Alongside an internal psychological approach valuing self-kindness, self-compassion is further defined by the recognition of common humanity (as opposed to self-isolation) and the practice of mindfulness (in place of emotional over-identification) [28]. Measures of self-compassion include facets such as self-kindness, self-judgment, feelings of common humanity and isolation, and tendencies toward over-identification [29]. Similar to most mindfulness-based approaches, an openness toward all human experiences, both pleasant and unpleasant, is integral to the practice of self-compassion [8,20]. In summary, it is reasonable to assume that self-compassion may facilitate mindful behavior and non-judgment toward intrapersonal as well as interpersonal experience.

### 2.3. The Nature and Functions of Self-Talk

Self-talk comprises covert or overt statements that are directed to the self as opposed to others [30]. There is evidence that self-talk includes both intrapersonal and interpersonal elements [30,31]. For example, internal dialogues are a type of intrapersonal communication in which the implementation of different voices and juxtaposed mental positions account for communication with not only the self, but also imagined communication with other figures [32]. As such, self-talk can be used to refer to any one of these things [33].

The positive and negative content of self-talk is a primary focus of research interest, particularly within clinical psychology [34]. The purpose of content-focused self-talk research is to explore how and why people talk to themselves and to gain a better understanding of the effects of different kinds of self-talk on the speaker [32]. Most self-talk scales differentiate categories of self-talk based on content or self-regulatory functions. For example, the Self-Talk Scale [33] designates four types of self-talk functions: self-criticism, self-reinforcement, self-management, and social assessment. These modes of intrapersonal communication refer to the assessment of negative events (e.g., when something bad happens), the assessment of positive events (e.g., in response to something good happening), general self-regulation behavior (e.g., figuring out what one needs to do), and the assessment of social interactions (e.g., replaying or rehearsing conversations with others), respectively.

In addition to self-talk content, research examines individual differences in the frequency of different kinds of self-talk. For example, Brinthaupt [35] reviews support for the cognitive disruption hypothesis, which postulates increased frequency of self-talk in response to negative self-related events that disrupt cognition (e.g., anxiety, obsessive-compulsive tendencies, and schizotypy). In other words, when people experience cognitive or emotional disruption, they tend to engage in more frequent self-talk to compensate for or resolve those disruptions. Taken together, theory and research on self-talk suggests the possibility of interesting relationships with trait mindfulness and self-compassion.

### 2.4. Possible Relationships among Trait Mindfulness, Self-Compassion, and Self-Talk

Research on trait mindfulness, self-compassion, and self-talk suggests a range of possible connections among these constructs. Whereas there has been some significant research on the links between mindfulness and self-compassion [7,19,20], much less is known about how self-talk relates to these variables. The possible interconnectivity between these constructs can be addressed in three parts: self-regulatory capabilities, attentional foundations, and the occurrence of disruptive or maladaptive cognitions.

There is good evidence to suggest that trait mindfulness, self-compassion, and self-talk play a significant role in psychological management and self-regulation. For example, Bishop et al. [1] posited that self-regulation could be a conditional variable for the effectiveness of mindfulness-based treatments. Self-regulatory disruptions are one of the proposed theoretical rationales for increases in self-talk frequency [25]. This notion is further backed by evidence that non-first-person self-talk (which is self-distancing) [36] enhances self-regulatory efforts. Additionally, interventions targeting the development of increased self-compassion have been shown to be just as effective as other techniques used to improve behavioral self-regulation [8].

Mindfulness, self-compassion, and self-talk are further bound by attentional underpinnings. Bishop et al. defined mindfulness as in part being “the self-regulation of attention” [1] (p. 233). They noted three attentional skills necessary for self-regulation: sustained attention, attention switching, and the inhibition of elaborative processing. Kabat-Zinn further suggested that mindful attention should include “an affectionate, compassionate quality within the attending, a sense of openhearted friendly presence and interest” [37] (p. 145), qualities consistent with most modern definitions of self-compassion. Researchers have also proposed that effective self-talk seems to be related to the regulatory focusing of an individual’s attention, particularly when presented with novel information [30,38]. The elaborative processing aspect of attention in particular appears to involve intrapersonal communication.

Disruptive or maladaptive cognitions is another area of possible overlap among trait mindfulness, self-compassion, and self-talk. Ruminative thoughts and heightened emotional reactivity, which are commonly associated with mood, anxiety, and obsessive-compulsive disorders, have been linked to less mindful engagement and lower levels of self-compassion [1,2,4,22]. A review by Gu et al. [39] found that rumination and worry mediate the psychological outcomes of mindfulness-based interventions. Most of the features of anxiety contrast with the primary mindfulness concepts: attention and openness to experience, non-elaborative thought patterns, decentering from self-derogatory thoughts, and avoidance of rumination [1,4]. Repetitive thinking patterns observed in rumination have also been shown to mediate the relationship between self-compassion and depressed and anxious feelings [22]. In contrast to the experiences of anxious individuals, people with higher trait mindfulness scores are less reactive to threats, as shown by brain pattern data [12]. There are also clear ties between anxiety and other neurotic disorders which have been associated with a higher frequency of self-critical self-talk [33] and lower levels of mindfulness [40].

In their discussion of the self-talk and mindfulness relationship, Leary and Tate argued: “In part, mindful attention is achieved by reducing one’s inner self-talk. Only by quieting self-chatter—the running flow of mental commentary, thoughts about the past and future, self-evaluations, judgments, and other extraneous reactions—can people remain highly attuned to their present experience” [9] (p. 252). Rather than engaging in ruminative pathological thought patterns, mindfulness involves momentary, non-elaborative experience in the mind and body [1,41].

### 2.5. Derivation of Hypotheses

Past theory and research suggest several hypotheses about the relationships between trait mindfulness, self-compassion, and self-talk. We first predicted that positive self-talk would be significantly and positively correlated with trait mindfulness, particularly with the non-judging, non-reacting, and decentering facets of mindfulness. Support for this hypothesis comes from the cognitive and emotional benefits of both positive self-talk and mindful practice. Positive affect has been shown to be predictive of mindfulness levels [18] and trait mindfulness has been tied to reductions in anxiety [10]. In a similar vein, self-talk has been shown to have a positive effect on self-confidence and to reduce performance anxiety [38].

Secondly, we expected positive self-talk to be significantly and positively correlated with self-compassion. Motivational self-talk has been shown to induce positive affective states and more generally increase self-compassion [42]. Research has also identified self-compassion as a theoretical immunization strategy for combatting patterns of maladaptive intrapersonal communication, such as depressive rumination and anxious worrying [22].

For our third hypothesis, we predicted that negative self-talk would be significantly and negatively correlated with both trait mindfulness and self-compassion. Higher mindfulness scores, particularly those associated with the describing, non-judging, and non-reacting facets of mindfulness, have been found to be negatively correlated with emotional disturbances [12] and negative emotions [18]. Negative (especially self-critical) self-talk tends to reflect disruptive or maladaptive cognitions, and a greater frequency of this kind of self-talk should be associated with lower trait mindfulness (specifically the facets of acting with awareness, non-judging, and non-reactivity) as well as self-compassion.

Finally, we expected that mindfulness practice would moderate the relationship between self-talk and self-compassion as they individually relate to trait mindfulness. According to Leary and Tate’s proposal, mindful development should be associated with a diminishing in self-communication with the adoption of a decentered, non-judgmental perspective [9]. This is further supported by evidence that self-regulatory disruptions (which can be attended to with mindful practice) are associated with increased self-talk frequency [35]. As such, we expect that more experience with mindful practice will correspond with more significant connections in the relationships among self-talk, self-compassion, and trait mindfulness when compared to those with a lesser history of mindful practice.

## 3. Materials and Method

### 3.1. Participants

The sample comprised 342 individuals recruited from both a large public southeastern US university’s psychology research pool (*n* = 293) and social media posting via Facebook (*n* = 49). We informed participants that the project was designed to help us explore the relationship between self-talk, mindfulness, and other related variables and that they would respond to a survey about their personal experiences with aspects of both self-talk and mindfulness. Research pool participants received course credit for their participation in this study. The sample included 222 women and 108 men, with seven participants identifying as non-binary and five participants not responding. The ethnic representation of the sample was: Caucasian (*n* = 223; 65.2%), African American (*n* = 49; 14.3%), Hispanic (*n* = 16; 4.7%), Asian (*n* = 8; 2.3%), mixed ethnicity (*n* = 24; 7.1%), and other (*n* = 20; 5.8%). Most participants (*n* = 288; 84%) were between the ages of 18 and 24 years old. To be eligible to participate, individuals had to be at least 18 years of age. Experience with mindful practice was not required for participation.

### 3.2. Measures

To test our hypotheses, we administered two common measures of trait mindfulness that captured different facets of the construct. We also used two frequently used self-talk measures, one that focused on the functions served by self-talk and the other that focused on the positive and negative content of self-talk. We also had participants complete a popular short form measure of self-compassion and provide demographic information.

*15-Item Five Facet Mindfulness Questionnaire (FFMQ-15)* [16]. The FFMQ is a frequently used measure of trait mindfulness [18]. The 15-item version of this scale assesses five facets of general mindfulness: *observing* (e.g., “I notice the smells and aromas of things”), *describing* (e.g., “I am good at finding words to describe my feelings”), *acting with awareness* (e.g., reverse-scored “I find myself doing things without paying attention”), *nonjudging of inner experience* (e.g., reverse-scored “I think some of my emotions are bad or inappropriate and I should not feel them”), and *nonreactivity to inner experience* (e.g., “I perceive my feelings and emotions without having to react to them”). Three items are associated with each of the five subscales (in which higher scores indicate more trait mindfulness). Each item is rated on a 5-point scale (1 = *never or very rarely true*, 5 = *very often or always true*). Prior research has established that both the long and short form of the FFMQ measure highly similar constructs and have a significant loading on an overall mindfulness factor [43]. Total and subscale internal consistencies have been reported as adequate for populations with therapeutic mindfulness-based intervention experience (i.e., 0.69 to 0.83) and without similar experience (i.e., 0.64 to 0.80) [43].

*Trait Toronto Mindfulness Scale (TMS-T)* [17]. The TMS-T is a 13-item trait mindfulness scale adapted from the Toronto Mindfulness Scale [44], which measures state mindfulness. The TMS-T assesses two factors of mindfulness: *decentering* and *curiosity*. Decentering emphasizes a distanced awareness of everyday experience (e.g., “I experience myself as separate from my changing thoughts and feelings”). The curiosity factor specifically reflects an inquisitive quality relevant to awareness in the present moment (e.g., “I am curious about what I might learn about myself by taking notice of how I react to certain thoughts, feelings, or sensations”). Six items are associated with the curiosity subscale and seven items are associated with the decentering subscale. Each item is rated on a 5-point scale (1 = *not at all*, 5 = *very much*), with high scores indicating a more mindful disposition. The Trait TMS uses the same format as the original measure and has been found to have good internal consistency reliability (i.e., 0.91 for *curiosity* and 0.85 for *decentering*), as reported by Davis et al. [17]. The TMS-T also was reported to have convergent validity with six other mindfulness measures, including the FFMQ [17].

*Self-Talk Scale (STS)* [33]. The STS is a 16-item scale measuring self-talk frequency across four function subscales: *social assessment* (refers to an individual’s social encounters; e.g., “I talk to myself when I’m imagining how other people respond to things I’ve said”), *self-criticism* (assesses negative events; e.g., “I talk to myself when I should have done something differently”), *self-reinforcement* (assesses positive events; e.g., “I talk to myself when something good has happened to me”), and *self-management* (refers to general self-regulation; e.g., “I talk to myself when I need to figure out what I should do or say”). Four items are associated with each of the subscales. Items are rated on a 5-point frequency scale (1 = *never*, 5 = *very often*). Scores can be summed at both the subscale and overall level, with higher scores indicating more frequent self-talk. Prior research has supported the use of the STS as a unidimensional measure of self-talk frequency [45]. This measure also has been shown to have good test–retest reliability for total scores [33] and good internal consistency for both total and subscale scores (i.e., 0.85 to 0.94) [35].

*Automatic Thoughts Questionnaire-Revised (ATQ-R)* [34]. The ATQ-R is a 40-item scale measuring positive and negative automatic self-statements related to depression (negative sample items: “I feel like I’m up against the world,” “I am a failure,” and “Nothing feels good anymore”; positive sample items: “I’m proud of myself,” “I can accomplish anything,” and “I feel good”). Thirty items measure the frequency of negative self-statements, and 10 items measure the frequency of positive self-statements. Respondents rate the items with a 5-point frequency scale (1 = *not at all*, 5 = *all the time*), in which higher scores indicate thoughts occurring more frequently in the previous week. This measure has proven reliable in discriminating depressed and non-depressed individuals in clinical and subclinical populations and retains a high internal consistency (i.e., coefficient alpha of 0.91) when dealing with non-clinical populations [46].

*Self-Compassion Scale—Short Form (SCS-SF)* [29]. The SCS-SF is a 12-item scale assessing six components of self-compassion: *self-kindness* (e.g., “I’m kind to myself when I’m experiencing suffering”), *self-judgment* (e.g., “When times are really difficult, I tend to be tough on myself”), *common humanity* (e.g., “When I feel inadequate in some way, I try to remind myself that feelings of inadequacy are shared by most people”), *isolation* (e.g., “When I fail at something that’s important to me I tend to feel alone in my failure”)*, mindfulness* (e.g., “When something upsets me I try to keep my emotions in balance”), and *over-identification* (e.g., “When I’m feeling down I tend to obsess and fixate on everything that’s wrong”). Two items are associated with each of these components. Each item is rated on a 5-point frequency scale (1 = *almost never*, 5= *almost always*). Comparative research by the developers has shown a very strong correlation with the long form for overall self-compassion scores. This measure has demonstrated high overall internal consistency (i.e., coefficient alpha of 0.86) and variable subscale level internal consistency (i.e., 0.54 to 0.75). The authors suggest that subscale divisions only be made when using the short form of the SCS if the derived information is crucial to the study at hand [29]. Evaluations of subscale level internal consistency for the current study showed similar variability (i.e., 0.57 to 0.78).

*Demographic Questionnaire.* We included a brief demographic questionnaire at the end of survey. These items assessed age, ethnicity, gender identity, and experience with mindfulness practice. For the latter, participants first indicated whether they had ever heard of the term “mindfulness” (yes/no). If they answered yes, we asked them to indicate if they had ever tried to practice mindfulness (yes/no). If yes, they then indicated how many days each week they practice mindfulness or mindfulness-based meditation (0–7).

### 3.3. Procedure

Participants completed an online survey through the Qualtrics system (www.qualtrics.com (accessed on 1 August 2022)). The order of presentation of the main measures was randomized across participants, with the brief demographic survey always appearing at the end of the survey. Informed consent and assurances of anonymity were integrated at the beginning of the survey and a debriefing statement appeared at the conclusion of the study. The project received approval from the university’s IRB prior to starting.

## 4. Results

### 4.1. Descriptive Statistics

The means, standard deviations, and internal consistency values for the mindfulness, self-compassion, and self-talk measures are reported in Table 1. For each variable, scores were comparable to previously published norm ranges provided by measures’ developers. As multiple measures were selected to represent the single variables of trait mindfulness and self-talk, some convergence was expected between shared constructs. As seen in Table 2, there were some significant correlations between the self-talk measures. Self-managing self-talk, though used as a positive self-talk measure in this research, was unexpectedly observed to be significantly positively correlated with *negative* ATQ scores, but not with positive ATQ scores. Subscale-level correlational analyses for the mindfulness measures are reported in Table 3. As some authors chose to provide normative scores for groups with and without mindfulness experience, it is important to address the corresponding make up of our sample. Though 94% of participants (*n* = 322) indicated that they have heard of mindfulness, only 54% of respondents (*n* = 184) reported having some experience with mindful practice, and most indicated that they currently practice mindfulness only one to two days a week (*M* = 1.73, *SD* = 1.75).

### 4.2. Test of Hypotheses

Hypothesis 1 sought to explore the nature of the relationship between positive self-talk and trait mindfulness, in which positive self-talk was expected to be significantly and positively correlated with trait mindfulness, and more particularly, the non-judging, non-reacting, and decentering facets of mindfulness. Zero-order correlations were used to gauge the strength and direction of the relationship between these variables (see Table 4).

Some positive correlations were observed between trait mindfulness and self-reinforcement, as well as between trait mindfulness and automatic positive self-statements. Specifically, self-reinforcing self-talk was significantly positively correlated with the acting with awareness and non-judging subscales of the FFMQ, in addition to the curiosity and decentering subscales of the TMS-T. Though they were observed to be negatively correlated with the describing facet of the FFMQ, automatic positive self-statements (ATQ-R) were further found to be positively correlated with the acting with the awareness, non-reactivity, and non-judging subscales of the FFMQ, in addition to both the curiosity and decentering subscales of the TMS-T. Consistent positive relations were not evident when considering the relationship between self-managing self-talk and trait mindfulness. Self-managing self-talk was instead negatively correlated with the non-judging subscale of the FFMQ and positively correlated with the curiosity subscale of the TMS-T. This correlation was in the opposite direction of those observed for STS self-reinforcement and ATQ-R positive self-statements. Still, there was overall moderate support for this hypothesis.

Hypothesis 2 proposed a significant positive correlation between positive self-talk and self-compassion scores. We again used zero-order correlations to determine the nature of the relationship between these variables (see Table 5). STS self-reinforcement and automatic positive self-statements were significantly positively correlated with overall self-compassion scores. STS self-management was negatively correlated with composite self-compassion scores. Considering possible explanations for this discrepancy, it is possible that for the present sample, self-managing self-talk was more corrective and strict than reaffirming and constructive. Still, given these data, there was good support for our hypothesis.

Hypothesis 3 investigated the relationship between negative self-talk and trait mindfulness, in which negative self-talk was expected to be negatively correlated with trait mindfulness (particularly the facets of acting with awareness, non-judging, and non-reactivity). We used zero-order correlations to investigate this relationship (Table 6). Self-critical self-talk was negatively correlated with the acting with awareness and non-judging subscales of the FFMQ and the decentering subscale of the TMS-T. Self-critical self-talk was also significantly positively correlated with the curiosity subscale of the TMS-T and the describing facet of the FFMQ. Social assessing self-talk was correlated negatively with the acting with awareness and non-judging subscales of the FFMQ, in addition to being positively correlated with the curiosity subscale of the TMS-T. Automatic negative self-statements were correlated negatively with the acting with awareness, non-reactivity, and non-judging subscales of the FFMQ. Negative ATQ-R scores were also positively correlated with the describing facet of the FFMQ and the curiosity subscale of the TMS-T. These results show moderate support for Hypothesis 3.

According to Hypothesis 4, experience with mindfulness practice was predicted as a moderating variable between self-talk and trait mindfulness (H4a), as well as between self-compassion and trait mindfulness (H4b). We analyzed these relationships using linear regression models which contained moderator variables created as interaction terms between the independent variables (self-compassion and self-talk) and the moderator (mindfulness practice, i.e., “How many days each week do you practice mindfulness or mindfulness-based meditation?”). These analyses (see Table 7) did not show any mindfulness experience moderating effects on self-compassion or self-talk frequency for trait mindfulness.

## 5. Discussion

The purpose of this research was to investigate the nature of the relationships among mindfulness, self-compassion, and self-talk. To evaluate these relationships, we generated hypotheses with attention to content differences in self-talk. All of the hypotheses were at least partially supported by the data, consistent with the posited theoretical framework of an existing relationship between these variables.

According to the results of the correlational analyses, positive self-directed communication (self-reinforcing self-talk and automatic positive self-statements) were associated with several aspects of trait mindfulness. Prior research and the existing literature support this finding, in which positive emotional states have historically been associated with positive mental imagery and internal communication [47] and trait mindfulness has been shown to predict positive emotional states [15].

Most striking in the consideration of the results associated with “positive,” self-oriented communication, however, was the role of self-managing self-talk. This particular facet of self-talk was negatively correlated with non-judging of inner experiences and positively correlated with curiosity. A possible explanation for this discrepancy is that self-management involves some measure of self-judgment and internalization. Though previous research using the STS has shown that positive thoughts have been associated with self-managing self-talk [33], the STS does not assess affective thought-content discrepancies as we examined in this study. Baer et al. [48] posited that self-talk, and self-focused attention more generally, can be maladaptive and associated with negative emotional responding, which may explain the negative correlation. These results further highlight the possibility that self-talk content and the use of self-talk play an important role in mindful regulation.

Given the results of the hypothesis testing accounted for in the present research, it seems to us that delineations of *positive* and *negative* as the ways in which self-talk relates to mindfulness may be an oversimplification. It is quite possible, given the literature, that self-talk and mindfulness are measuring something outside of the established factorial purview of the measures from which they were derived. The observing facet of the FFMQ remained largely unaffiliated with other variables in the present research. Baer et al. [48] provide further evidence for why it may not be associated with the self-talk and self-compassion variables, noting that the observing facet in the original FFMQ was found to be positively correlated with several maladaptive constructs, including thought suppression, dissociation, absentmindedness, and other associated psychological symptoms. The results reported here also support previous research by Baer et al. [16,48] which suggests removing the observing facet and instead using a four-factor hierarchical model of overall mindfulness when attempting research with non-meditating populations.

As was previously reported, only roughly half of participants had experience with mindfulness and of that half, very few engaged in consistent mindfulness practice. Scores between those with and without mindfulness practice did not different substantially, possibly due to the restricted range of this variable. This may have been a complicating factor for the moderation analysis, in which mindfulness practice was not found to moderate the relationships between self-talk frequency or self-compassion as they related to trait mindfulness. Future research in this vein could benefit from the inclusion of a population with more regular mindfulness practice.

Research has supported some connections through self-regulation abilities, attentional underpinnings, and reactivity to negative symptoms associated with maladaptive cognitions [1,39]. The current findings provide additional insights into these connections. First, our results showed a positive connection between positive self-talk as it relates to trait mindfulness and self-compassion. This result implies a potential benefit of positive intrapersonal communication in the development of mindfulness and a self-compassionate attitude. However, we also found that increases in self-managing self-talk were not related to positive self-communication and self-compassion. Though these results are consistent with previous research showing that self-regulatory disruptions are associated with increases in self-talk frequency [35], they suggest that the *content* of the intrapersonal communication is essential in efforts to establish any psychological benefits.

Prior research has shown that as mindfulness increases, emotional disturbances and negative emotions decrease [12,18]. We found, consistent with the present research, a significant negative correlation between negative self-talk and trait mindfulness, with particular respect to the components of non-judgment, non-reactivity, and awareness (which are traditionally integral to a mindful disposition). These results provide support for the idea of a connection between psychological disorders associated with negative self-talk (e.g., anxious worries and depressive rumination) [33] and lower levels of mindfulness [40]. Leary and Tate [9] argued that the development of mindful attention supposes a reduction in “self-chatter” and critical self-judgment. Our results support this argument with some qualifications. It appears that the self-regulatory aspects of self-talk are not so much related to the suppression of thoughts or intrapersonal communication as they are the building of a mindful disposition and self-compassionate attitude.

### 5.1. Limitations of the Research

There were a variety of limitations to this study. Based on the demographic statistics, the sample appeared lacking in diversity. Most of the individuals who participated were undergraduate students from the same southeastern U.S. university, in which 84.2% of the participants were between the ages of 18 and 24 years old. Additionally, participants were predominantly white (65.2%), which indicates a group not entirely representative of the general population. This may impact the generalizability of these results to other ethnic groups and age demographics.

The present sample may also have been affected by complications due to the coronavirus pandemic. Due to pandemic restrictions, on-campus student participants not enrolled in the university’s psychology research pool were unable to participate, greatly limiting recruitment efforts for the study. This may have played a significant role in the aforementioned lack of sample diversity. The stress and emotionality associated with navigating a global pandemic may have further influenced the mental states of participants, lending to response styles which may not be entirely representative of the sample under more ordinary circumstances. The use of an electronic survey method due to pandemic restrictions may also have resulted in variance between intercorrelated variables as a product of a shared method of presentation.

Further, the study model was somewhat basic, with particular consideration of the moderation analysis and the potential effects of participant mindfulness practice on the target variables. Future research may benefit from additional attention given toward the methods of mindful practice engaged in by participants (e.g., breathing exercise, yoga mindfulness, walking mindfulness). While the present research determined experience with mindfulness as based upon self-reported engagements in mindful practice each week, the inclusion of supplemental questions regarding the nature and length of participants’ personal practice could shed more light on the relationship mindfulness has to other metacognitive variables.

Using short forms of several of the assessment tools in this study also limited the present research. Results derived from the Self-Compassion Scale were most predominantly affected. The creator of the measure advises against examination of subscales when using the short form [29], so there was potential additional information which was lost. This information could have proven useful in drawing further conclusions about the specifics of the relationship between self-compassion and the other variables of interest.

### 5.2. Implications for Future Research

Future research should continue to explore associations between self-talk frequency and other cognitive variables, as were examined in this study with self-compassion and trait mindfulness. Although experience with mindfulness practice did not help to predict correlations between intrapersonal communicative variables and trait mindfulness, there is still support for a connection between self-oriented cognition and other cognitive variables which have been historically observed to affect treatment outcomes. Cognitive disruptions, such as anxiety and obsessive-compulsive tendencies, which were previously noted to be associated with increased self-talk and decreased self-regulation [35], are commonly treated with interventions which target the negative thought patterns produced by the cognitive disruptions. Moreover, this treatment approach is often accomplished by incorporating mindfulness-based training. It would be worthwhile to further investigate this relationship in such a way that changes in self-talk frequency and other self-oriented variables might be observed to directly correspond with changes in individual mindfulness.

Additionally, future studies could explore the possibility of using mindfulness interventions to better target certain established categories of self-talk and self-talk content to contribute to the treatment of mental health disorders and ruminative coping styles. Exploring the notion of self-talk as a transdiagnostic component could be extremely helpful in the case of treatment planning. In taking steps to identify what types of negative self-communication a client uses, service providers would be able to better serve the client by adapting the therapeutic intervention to the relevant self-communication areas in which work is needed to see long-term psychological improvement. Previous research has already shown that interventions which promote mindful self-compassion can be especially beneficial in the management of chronic symptoms of posttraumatic stress [23]. It appears that these kinds of interventions could be an effective way to combat the self-communicative maintenance component of many recurrent psychological disorders.

Expanding upon this idea, there is a possibility that the exploration of self-talk style recognition could be more generally employed in the selection of a mindfulness intervention style most appropriate to a client’s patterns of self-communication. Research by Woods and Proeve [24], for instance, sought to establish a relationship between mindfulness and self-compassion as they relate to shame-proneness. They found that self-compassion levels were predictive of proneness to feelings of shame. However, the same relationship was not observed between shame-proneness and mindfulness. This suggests that individuals who are more prone to feelings of shame may benefit from a self-compassion-focused intervention, rather than one targeted more particularly to the development of a mindful disposition.

### 5.3. Conclusions

In summary, the present findings suggest that a relationship does exist between trait mindfulness and internal variables (e.g., intrapersonal communication and a self-compassionate attitude) which affect emotional and cognitive regulation. Furthering our knowledge of the range of ways in which these metacognitive factors can interact with and affect psychological outcomes is worth prioritizing in the field to assist with the development of personalized intervention methods designed to combat psychopathology. As previous research by Hölzel et al. purports, mindfulness practice alone engages several cognitive mechanisms which result in the reduction in negative symptomology associated with psychopathology [10]. To better describe how this notion fits into the foundational theoretical framework proposed in the present study, it is important not only to promote mindful engagement, but to also investigate interactions between factors which may encourage or inhibit the development of a mindful disposition that is resistant to cognitive disruption. The present findings contribute to a more fleshed out understanding of the intertwining of inner experiences which may impact overall treatment efficacy and efficiency.

While different styles of MBI are variably effective for different clientele (as well as different disorders), exploring other influential correlates (even beyond those considered here with self-talk frequency and self-compassion) could be helpful in selecting intervention styles which best match the individual case of each client. Again, this could be especially effective in assisting clients struggling with cognitive disruptions characterized by internalization, heightened self-focus, and maladaptive self-communication.

## Figures and Tables

**Table 1 behavsci-12-00300-t001:** Descriptive statistics for major measures.

Questionnaire	Number of Items	*n*	Mean	Std. Deviation	Cronbach’s Alpha
15-Item Five Facet Mindfulness Questionnaire (FFMQ-15)	15	336	45.04	3.98	0.723
Describing	3	341	9.13	1.49	0.814
Observing	3	340	8.65	1.74	0.572
Acting with Awareness	3	340	9.07	1.38	0.718
Non-Judging	3	340	9.31	1.70	0.810
Non-Reactivity	3	339	8.89	2.53	0.663
Trait Toronto Mindfulness Scale (TMS-T)	13	334	40.17	8.33	0.825
Curiosity	6	338	20.72	5.30	0.869
Decentering	7	336	19.41	4.68	0.695
Self-Talk Scale (STS)	16	329	59.25	9.98	0.864
Self-Criticism	4	336	14.16	3.83	0.841
Self-Reinforcement	4	337	13.51	3.62	0.838
Self-Management	4	334	16.06	2.97	0.755
Social Assessment	4	336	15.49	3.57	0.827
Self-Compassion Scale—Short Form (SCS-SF)	12	337	33.70	8.39	0.850
Automatic Thoughts Questionnaire-Revised (ATQ-R)	40	317	98.23	23.99	0.968
Positive	10	330	29.96	8.13	0.888
Negative	30	324	68.46	27.50	0.971

*Note.* Possible FFMQ-15 subscale scores range from 3 to 15. Possible TMS-T Curiosity subscale scores range from 0 to 24. Possible TMS-T Decentering subscale scores range from 0 to 28. Possible STS subscale scores range from 4 to 20. Possible SCS-SF scores range from 12 to 60. Possible ATQ-R positive scores range from 10 to 50. Possible ATQ-R negative scores range from 30 to 150.

**Table 2 behavsci-12-00300-t002:** Pearson correlations between self-talk measures.

	ATQ Negative	ATQ Positive
STS Total	0.302 **	−0.073
STS Self-Criticism	0.537 **	−0.416 **
STS Self-Reinforcement	−0.213 **	0.482 **
STS Self-Management	0.230 *	−0.094
STS Social Assessment	0.298 **	−0.161 *

*Note*. *N =* 319 to 336. STS = Self-Talk Scale; ATQ = Automatic Thoughts Questionnaire; * *p* < 0.05, ** *p* < 0.01.

**Table 3 behavsci-12-00300-t003:** Pearson correlations between mindfulness measures.

	TMS Curiosity	TMS Decentering
FFMQ Composite	0.105	0.298 **
FFMQ Describing	−0.045	−0.009
FFMQ Observing	0.142 **	0.077
FFMQ Acting with Awareness	−0.079	0.036
FFMQ Non-Judging	−0.057 *	0.110 *
FFMQ Non-Reactivity	0.172 **	0.322 **

*Note. N* = 334 to 336. FFMQ = Five Facet Mindfulness Questionnaire; TMS = Toronto Mindfulness Scale; * *p* < 0.05, ** *p* < 0.01.

**Table 4 behavsci-12-00300-t004:** Pearson correlations between positive self-talk and trait mindfulness.

	STS Self-Reinforcement	STS Self-Management	ATQ Positive
FFMQ Composite	0.218 **	−0.020	0.297 **
FFMQ Describing	−0.061	0.085	−0.190 **
FFMQ Observing	0.083	0.047	0.068
FFMQ Acting with Awareness	0.199 **	−0.078	0.214 **
FFMQ Non-Judging	0.174 *	−0.162 *	0.303 **
FFMQ Non-Reactivity	0.101	0.04	0.213 **
TMS Curiosity	0.279 **	0.398 **	0.234 **
TMS Decentering	0.218 **	0.013	0.308 **

*Note. N* = 317 to 336; STS = Self-Talk Scale; ATQ = Automatic Thoughts Questionnaire; FFMQ = Five Facet Mindfulness Questionnaire; TMS = Toronto Mindfulness Scale; * *p* < 0.05, ** *p* < 0.01.

**Table 5 behavsci-12-00300-t005:** Pearson correlations between positive self-talk and self-compassion.

	SCS Composite Score
STS Self-Reinforcement	0.334 **
STS Self-Management	−0.206 **
ATQ Positive	0.601 **

*Note*. *N* = 326 to 337; STS = Self-Talk Scale; ATQ = Automatic Thoughts Questionnaire; SCS = Self-Compassion Scale; ** *p* < 0.01.

**Table 6 behavsci-12-00300-t006:** Pearson correlations between negative self-talk and trait mindfulness.

	STS Self-Criticism	STS Social Assessment	ATQ Negative
FFMQ Composite	−0.200 **	−0.107 *	−0.194 **
FFMQ Describing	0.124 *	0.111 *	0.293 **
FFMQ Observing	−0.011	0.048	0.120
FFMQ Acting with Awareness	−0.185 **	−0.179 *	−0.312 **
FFMQ Non-Judging	−0.288 **	−0.203 **	−0.337 **
FFMQ Non-Reactivity	−0.079	−0.026	−0.172 **
TMS Curiosity	0.213 *	0.325 **	0.088
TMS Decentering	−0.099	−0.004	0.011

*Note. N* = 319 to 336; STS = Self-Talk Scale; ATQ = Automatic Thoughts Questionnaire; FFMQ = Five Facet Mindfulness Questionnaire; TMS = Toronto Mindfulness Scale; * *p* < 0.05, ** *p* < 0.01.

**Table 7 behavsci-12-00300-t007:** Moderation effects of experience with mindfulness practice predicting trait mindfulness for self-talk and self-compassion. Coefficients ^a^.

	Unstandardized Coefficients		Standardized Coefficients		
Model	B	Std. Error	Beta	*t*	Sig.
(Constant)	50.5	2.81		17.93	<0.001
Practice	−0.987	1.20	−0.403	−0.826	0.410
STS Composite	−0.114	0.045	−0.251	−2.51	0.013
STS Composite × Practice	0.031	0.019	0.808	1.65	0.102
(Constant)	36.2	1.52		23.8	<0.001
Practice	1.01	0.650	0.422	1.56	0.121
SCS Composite	0.228	0.045	0.457	5.07	<0.001
SCS Composite × Practice	−0.010	0.017	−0.186	−0.626	0.532

*Note*. Practice = “How many days each week do you practice mindfulness or mindfulness-based meditation?” (0–7); STS = Self-Talk Scale; SCS = Self-Compassion Scale; ^a^ Dependent Variable: Five Facet Mindfulness Questionnaire (FFMQ) Composite Score.

## Data Availability

Not applicable.

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
