# Peer review of "Trait Mindfulness, Self-Compassion, and Self-Talk: A Correlational Analysis of Young Adults"

_behavsci, 2022, doi:10.3390/bs12090300_

Round 1
Reviewer 1 Report
Thank you so much for giving me an opportunity to review this manuscript. The manuscript examined the relationship between trait mindfulness, self-compassion, self-talk, and mindfulness practice. It is a well-structured manuscript, easy to follow and understand. Following are a few recommendations that might help to improve the manuscript.
1. The introduction section indicated the literature gap but it did not highlight why it is important to address this particular literature gap. I would suggest to highlight why it is important to examine the relationship among trait mindfulness, self-compassion, and self-talk? The broader domain or context of the study is not clear as well.
2. Fourth hypothesis analysed the role of mindfulness practices in moderating the relationship between self-talk and trait mindfulness as well as the relationship between self-compassion and trait mindfulness. Experience with mindfulness practices is measured as the number of times the respondent practice mindfulness in a week. The study did not consider the nature of mindfulness practices that research participants adopted like breathing exercise, yoga mindfulness, walking mindfulness etc? What is duration of their weekly practices? How many years of mindfulness practices experience they have? In addition,"only 54% of respondents (n = 184) reported having some experience with mindful practice, and most indicated that they currently practice mindfulness only one to two days a week". It is difficult for me to understand how can you examine the role of mindfulness practice without considering the nature of practices or experiences. Please clarify.
3. It is quite unusual to see the definition of “self-managing self-talk” in the discussion section. I would suggest to relocate it from discussion to the introduction/literature review section.
4. The practical and research implications of study is missing from the manuscript. Please discuss how the research findings add value to the existing literature or theory and practice. For instance, it says “negative self-talk variables were found to partially support the hypothesis, where they indicated negative correlations with trait mindfulness”. The theoretical contributions of this research findings is not clear. Please explain in the discussion section the contextual and theoretical background of this research and discuss how this research study add value to the existing literature and practice.
Author Response
Reviewer #1:
Comments and Suggestions for Authors
- The introduction section indicated the literature gap but it did not highlight why it is important to address this particular literature gap. I would suggest to highlight why it is important to examine the relationship among trait mindfulness, self-compassion, and self-talk? The broader domain or context of the study is not clear as well. – We have re-organized and further developed the Introduction section to indicate the importance of current research in relation to the outcomes of psychological interventions (lines 37-50).
- Fourth hypothesis analysed the role of mindfulness practices in moderating the relationship between self-talk and trait mindfulness as well as the relationship between self-compassion and trait mindfulness. Experience with mindfulness practices is measured as the number of times the respondent practice mindfulness in a week. The study did not consider the nature of mindfulness practices that research participants adopted like breathing exercise, yoga mindfulness, walking mindfulness etc? What is duration of their weekly practices? How many years of mindfulness practices experience they have? In addition, "only 54% of respondents (n = 184) reported having some experience with mindful practice, and most indicated that they currently practice mindfulness only one to two days a week". It is difficult for me to understand how can you examine the role of mindfulness practice without considering the nature of practices or experiences. Please clarify. – Good point; we have addressed this issue in the limitations section of the Discussion (lines 501-509).
- It is quite unusual to see the definition of “self-managing self-talk” in the discussion section. I would suggest to relocate it from discussion to the introduction/literature review section. – Line in question removed from Discussion section. Information added in Intro/Lit Review briefly defining the types of self-talk as described using the STS (lines 121-126).
- The practical and research implications of study is missing from the manuscript. Please discuss how the research findings add value to the existing literature or theory and practice. For instance, it says “negative self-talk variables were found to partially support the hypothesis, where they indicated negative correlations with trait mindfulness”.The theoretical contributions of this research findings is not clear. Please explain in the discussion section the contextual and theoretical background of this research and discuss how this research study add value to the existing literature and practice. – Thank you for the recommendation! The Discussion section is reorganized and edited to highlight the ways in which the current research confirms or relates to relevant theories and establishes a working relationship between the target variables that was not previously investigated (lines 416-482). We discuss practical implications of the established theoretical framework in the "Implications for Future Research" section (lines 517-551).
Reviewer 2 Report
The manuscript, It's interest important and emerging research in behavioral sciences area. The findings are interesting as intentions leads to performance which is a new dimension. There are some issue that should be revised or included by the authors:• The introduction should be more clearly organized, highlighting the context in which the study was conducted and what is the added contribution of this research to what we already know.
• Study Objectives should be mentioned at the end of the introduction section.
• The authors should cite some recent studies to support the hypothesis.
• I will suggest considering the mediator variable. The current study model is very basic.
• What information was given to the participants regarding the nature of the study? Were questionnaires anonymous? Who recruited the participants, and how were they recruited?
• Sample items should be provided for every scale used, including information regarding the response scales, alphas, and descriptive statistics of the study variables.
• The discussion should be reorganized, avoiding repetitions of the results/hypotheses and rather explaining how this study contributes to the field.
• Please acknowledge more clearly limitations regarding the research design and report evidence to address the potential of common method variance.
• This section needs to more clearly connect the results to the existing research and theoretical framework.
• Another aspect that could be greatly improved is that the discussion of the results is disconnected from the literature review carried out.
• Finally, the conclusions are not properly tied to the results, and it is concluded on matters not empirically verified in the document.
• Manuscript references should be as per journal requirements.
• Manuscript proofreading is required.
Good Luck
Author Response
Reviewer #2:
Comments and Suggestions for Authors
• The introduction should be more clearly organized, highlighting the context in which the study was conducted and what is the added contribution of this research to what we already know. – We have reorganized Introduction and included information regarding relevance of current research to psychological interventions (lines 37-50).
- Study Objectives should be mentioned at the end of the introduction section. – Done (lines 47-49)
- The authors should cite some recent studies to support the hypothesis. – We included the most up-to-date citations we could fine that are relevant to our hypotheses.
- I will suggest considering the mediator variable. The current study model is very basic. – We have addressed some of the problems with the moderation analysis in the limitations section of the Discussion (line 501-509).
- What information was given to the participants regarding the nature of the study? Were questionnaires anonymous? Who recruited the participants, and how were they recruited? – This information appears in the participants (lines 220-232) and procedure (lines 314-320) sections.
- Sample items should be provided for every scale used, including information regarding the response scales, alphas, and descriptive statistics of the study variables. – This information is provided in the Method section and the first part of the Results.
- The discussion should be reorganized, avoiding repetitions of the results/hypotheses and rather explaining how this study contributes to the field. – We have reorganized and streamlined the Discussion section to avoid repetition of hypotheses. In addition to discussing the implication of the results, we also highlight the potential theoretical importance of the results (lines 416-482).
- Please acknowledge more clearly limitations regarding the research design and report evidence to address the potential of common method variance. – Good point; we have added a note about possible method variance issues with our design and results (lines 498-500).
- This section needs to more clearly connect the results to the existing research and theoretical framework. Another aspect that could be greatly improved is that the discussion of the results is disconnected from the literature review carried out. – New Conclusions section (lines 552-573) and reorganization of the Discussion incorporate responses to information described in the literature review and hypothesis setup (lines 416-482).
- Finally, the conclusions are not properly tied to the results, and it is concluded on matters not empirically verified in the document. – We added a separate Conclusions section at the end of the manuscript to summarize the overarching importance of the present research (lines 552-573).
- Manuscript references should be as per journal requirements. – We checked formatting and made some minor adjustments to spacing that was off in this section.
• Manuscript proofreading is required. – Done
Reviewer 3 Report
The study is methodologically rigorous, however, it presents an important limitation with respect to the representativeness of the sample, witch can be corrected by modifying the title of the manuscript, since the findings are presents in university students: "Trait Mindfulness, self-compassion and self talk: a correlational analysis in university students"
Author Response
Reviewer #3:
Comments and Suggestions for Authors
The study is methodologically rigorous, however, it presents an important limitation with respect to the representativeness of the sample, witch can be corrected by modifying the title of the manuscript, since the findings are presents in university students: "Trait Mindfulness, self-compassion and self talk: a correlational analysis in university students" – We decided to use “young adults” since 17% of the sample was comprised of non-university students recruited via social media posting.
Round 2
Reviewer 2 Report
The manuscript has been significantly improved by adding all those changes. I believe that the paper is now much more interesting to future readers and the quality of the presentation has been raised to a higher level. Therefore, I would like to congratulate the authors on their excellent work. Goodluck!
Reviewer 3 Report
I continue to think that the sample is not representative, however, the rest of the work is good and I think it contributes a body of research to an expanding theme